# Strain Sensor via Wood Anomalies in 2D Dielectric Array

**DOI:** 10.3390/nano11041022

**Published:** 2021-04-16

**Authors:** Rashid G. Bikbaev, Ivan V. Timofeev, Vasiliy F. Shabanov

**Affiliations:** 1Kirensky Institute of Physics, Federal Research Center KSC SB RAS, 660036 Krasnoyarsk, Russia; tiv@iph.krasn.ru (I.V.T.); shabanov@ksc.krasn.ru (V.F.S.); 2Siberian Federal University, 660041 Krasnoyarsk, Russia

**Keywords:** Wood anomalies, dielectric array, strain sensor, bioinspired structure

## Abstract

Optical sensing is one of many promising applications for all-dielectric photonic materials. Herein, we present an analytical and numerical study on the strain-responsive spectral properties of a bioinspired sensor. The sensor structure contains a two-dimensional periodic array of dielectric nanodisks to mimic the optical behavior of grana lamellae inside chloroplasts. To accumulate a noticeable response, we exploit the collective optical mode in grana ensemble. In higher plants, such a mode appears as Wood’s anomaly near the chlorophyll absorption line to control the photosynthesis rate. The resonance is shown persistent against moderate biological disorder and deformation. Under the stretching or compression of a symmetric structure, the mode splits into a couple of polarized modes. The frequency difference is accurately detected. It depends on the stretch coefficient almost linearly providing easy calibration of the strain-sensing device. The sensitivity of the considered structure remains at 5 nm/% in a wide range of strain. The influence of the stretching coefficient on the length of the reciprocal lattice vectors, as well as on the angle between them, is taken into account. This adaptive phenomenon is suggested for sensing applications in biomimetic optical nanomaterials.

## 1. Instruction

Sensing is one of the most important applications of plasmonic and fully dielectric periodic arrays. This is due to the unique optical properties of the lattice [1,2,3,4,5,6] and Fano resonances [7,8,9], or bound states, in the continuum [10,11,12], which occur in such structures. Many effects, e.g., the surface lattice resonance, are based on Wood’s anomalies discovered more than a century ago [13] and later interpreted by Rayleigh [14] and Fano [15]. Optical biosensing most often includes monitoring of the change in the refractive index during deposition of biological molecules onto the nanoparticle surface. In this case, a local increase in the refractive index near a nanoparticle is caused by the fact that the refractive index (RI) of most biological molecules is higher than that of water (n=1.33) and lies in the range of $1.45<n_{b}<1.55$. The sensitivity of such devices is determined by the shape of its constituent elements. Therefore, structural elements can be spheroids [1], disks [16], dimers [17], and flat/2D [18] or 3D clusters [19]. Changing the shape and material of nanoparticles, one can obtain ultra-high-Q resonances, which enhance the accuracy of biosensor tests. The drawback of such devices is the inability to determine an analyte itself, as they only respond to the changes in the refractive index.

The optical properties of two-dimensional periodic arrays can be controlled not only by changing the refractive index of a surrounding medium, but also by varying a grating pitch. This gave rise to another direction called strain sensing [20,21,22]. This method consists in estimating a shift of the resonance line wavelength upon straining the grating. In strain-gauge sensing, it is important to ensure a high Q factor and tunability of the resonance peak in a wide wavelength range. A narrow linewidth, which is important for signal resolution at low strains, can be provided by placing nanoparticles onto a flexible or plastic substrate, e.g., a polydimethylsiloxane (PDMS) film [18,23,24,25,26]. The high Young’s modulus of this material prevents the formation of tensile cracks. The use of elastic materials makes such structures more like soft matter, or, in other words, nature-like. Therefore, the processes occurring in living media can be qualitatively described using the data of investigations of two-dimensional periodic arrays. When designing such devices, it is important to take into account the uniformity of the refractive index of the environment. The refractive indices of the substrate and superstrate must be equal. Otherwise, the sensitivity of the device decreases dramatically [27].

In this study, we examined the spectral properties of a two-dimensional triangular grating consisting of dielectric nanodisks placed in a homogeneous medium to mimic the optical behavior of grana lamellae inside chloroplasts [28]. We explored the spectral properties of the structure at different grating pitches. Analytical expressions for describing the behavior of Wood’s anomalies in such media were derived. The analytical results are compared with the simulation obtained by finite difference time domain method. It is shown that the results obtained by two different approaches are in good agreement. The possibility of using such structures in the development of strain sensor was analyzed.

## 2. Description of the Structure

A schematic of a two-dimensional triangular grating consisting of nanodisks is shown in Figure 1a. The refractive index of the disk is $n_{d}=1.55+0.01i$ [28]. The structure is immersed in a homogeneous medium with a refractive index of $n_{ext}=1.33$. The disk radius and height are r=178 nm and h=200 nm, respectively. The computational domain periods along the *x* and *y* axes were $p_{x}=578$ nm and $p_{y}=1001$ nm, respectively. The described model allows us to distinguish between these cases and consider deformations along the x- and y- axes independently of each other. Moreover, independent deformation in two directions will simplify the analytical description of the structure.

To calculate the transmittance spectra of the investigated structure, we used the finite difference time domain (FDTD) method [29]. In contrast to other methods, e.g., the coupled dipole method, the mathematical model underlying the FDTD method does not use approximations and is accurate for classical electrodynamics. The method is based on the discretization of Maxwell’s equations written in differential form. The equations are solved by the finite difference method on two nested structured rectangular grids: one of which is designed to calculate electric fields, the other for magnetic fields. The method allows one to obtain the transmittance, reflectance, and absorptance spectra of a system and the field distribution within the computational domain in one calculation, taking into account both the geometric structure of the system under study and the optical properties of the materials used. The imposition of periodic boundary conditions and accounting for the symmetry of the unit cell of a periodic structure (Figure 1a) significantly accelerate the computations. For the structure under study, the periodic boundary conditions were applied along the x and y axes. We have limited our consideration to a circularly polarized plane wave incident normally at the two-dimensional structure.

## 3. Wood’s Anomalies in Two-Dimensional Periodic Array

The positions of Wood’s anomalies and, consequently, resonance lines of a two-dimensional periodic grating can be determined using the expression
(1)$$k^{\mathrm{WA}}=k^{\mathrm{in}}+G$$
where $k^{\mathrm{WA}}$ and $k^{\mathrm{in}}$ are the wave vectors of the incident and diffracting waves, respectively, and G is the reciprocal-lattice vector.

Initially, the symmetry of the triangular lattice coincides with the point group symmetry D${}_{6h}$ in Schoenflies notation, because disc-shaped grana obtain higher circular symmetry and can be substituted by dimensionless points. The group D${}_{6h}$ contains well-known subgroups C${}_{6h}$ and C${}_{6}$, associated with z-oriented rotational symmetry axis.

The investigated two-dimensional structure is described by the direct-lattice vectors (see Figure 1a):(2)$$a_{1}^{D_{6h}}=(\frac{1}{2},\frac{\sqrt{3}}{2})a,\;a_{2}^{D_{6h}}=(\frac{1}{2},-\frac{\sqrt{3}}{2})a$$
where *a* is the lattice period.

The vectors of the deformed lattice can be obtained by multiplying the undeformed vectors by the deformation matrix $\hat{X}$. As a deformation, we consider the case of stretching the lattice along the x axis. For this purpose, we introduce the strain coefficient $\gamma =p/p_{x}$:(3)$$a_{1,2}=\hat{X}a_{1,2}^{D_{6h}},\;\hat{X}=[\begin{array}{cc} \gamma & 0\\ 0 & 1 \end{array}].$$
As a result, the vectors of the deformed lattice take the form
(4)$$a_{1}=(\frac{\gamma }{2},\frac{\sqrt{3}}{2})a,\;a_{2}=(\frac{\gamma }{2},-\frac{\sqrt{3}}{2})a.$$
Note that when the lattice is deformed, the symmetry of the lattice D${}_{6h}$ breaks down to D${}_{2h}$, but when the period of $p_{x}$ is increased exactly by $\sqrt{3}$ times, the symmetry is upgraded to D${}_{4h}$ (see Figure 1a). The direct-lattice vectors of a two-dimensional square periodic structure have the form
(5)$$a_{1}^{D_{4h}}=(\frac{\sqrt{2}}{2},\frac{\sqrt{2}}{2})a^{\prime },\;a_{2}^{D_{4h}}=(\frac{\sqrt{2}}{2},-\frac{\sqrt{2}}{2})a^{\prime },$$
where $a^{\prime }$ is the lattice period. In our case, $a^{\prime }=\sqrt{3/2}a$. Additional stretching returns symmetry to D${}_{2h}$ until the period is further increased exactly to $p=3p_{x}$. At this point, the symmetry is restored once again to D${}_{6h}$, and both lattice periods $|a_{1,2}|$ are increased by $\sqrt{3}$ times:(6)$$a_{1}^{D_{6^{\prime }h}}=(\frac{3}{2},\frac{\sqrt{3}}{2})a,\;a_{2}^{D_{6^{\prime }h}}=(\frac{3}{2},-\frac{\sqrt{3}}{2})a.$$
Additional period increment gives D${}_{2h}$ symmetry.

In general, vectors (Equation 4) correspond to the reciprocal-lattice vectors:(7)$$b_{1}=(\frac{1}{\gamma },\frac{1}{\sqrt{3}})\frac{2\pi }{a},\;b_{2}=(\frac{1}{\gamma },-\frac{1}{\sqrt{3}})\frac{2\pi }{a}.$$
In Equation (Equation 1), the $k^{\mathrm{WA}}=\pm 2\pi n_{ext}\lambda$, $k^{in}=2\pi \sin\left(\theta \right)/\lambda$, and $G=ib_{1}+jb_{2}$, where θ is the angle of light incidence and *i*, *j* are integers which represent the order of the phase difference in $a_{1}^{D_{6h}}$ and $a_{2}^{D_{6h}}$ vectors directions. We assume that these indices take the values 0, −1, and +1, respectively. In case of normal incidence of light onto the structure θ=0, we obtain
(8)$$2\pi n_{ext}/\lambda =\left|ib_{1}+jb_{2}\right|.$$
The resonant wavelength λ is derived as
(9)$$\lambda =\frac{2\pi }{|ib_{1}+jb_{2}|}n_{ext}.$$
Using the vector sum and taking into account the Equation (Equation 7), we can determine the wavelength $\lambda_{x}$ of the resonant line in the case of deformation of the structure along the x axis:(10)$$\lambda_{x}=\frac{a}{\sqrt{\left(\frac{1}{\gamma^{2}}+\frac{1}{3})(i^{2}+j^{2}-2ij(\frac{\gamma^{2}-3}{\gamma^{2}+3})\right)}}n_{ext}.$$
It is important that this equation is derived taking into account the effect of coefficient γ not only on the reciprocal-lattice vector length, but also on the angle between the vectors. It can be seen in Figure 1a that $\cos\left(\alpha /2\right)=\gamma a/\left(2\right|a_{1}\left|\right)=\gamma /\left(\sqrt{3+\gamma^{2}}\right)$. Then, we have $\cos\alpha =2\cos^{2}\left(\alpha /2\right)-1=\left(\gamma^{2}-3\right)/\left(\gamma^{2}+3\right)$.

Similarly, we can obtain an expression for the positions of Wood’s anomalies in the case of straining the grating along the y axis:(11)$$\hat{Y}=[\begin{array}{cc} 1 & 0\\ 0 & \gamma \end{array}].$$
We should rewrite the direct-lattice vectors and draw the above-described conclusions. As a result, we obtain
(12)$$\lambda_{y}=\frac{a}{\sqrt{\left(1+\frac{1}{3\gamma^{2}})(i^{2}+j^{2}+2ij(\frac{3\gamma^{2}-1}{3\gamma^{2}+1})\right)}}n_{ext}.$$

Thus, Equations (Equation 10) and (Equation 12) can describe the positions of Wood’s anomalies under straining the grating along the *x* and *y* axes (see Figure 2).

Interestingly, upon straining the structure along the x axis, the positions of anomalies [1;0] and [0;1] change, while the wavelength of anomalies [−1;1] and [1;−1] remains invariable ($\lambda_{x}\left(i=-j=1\right)=const$). This is due to the fact that the deformation of the lattice along the x axis does not lead to a change in the distance between the layers of nanodisks. When the grating is strained in the orthogonal direction, we have a different picture, as, in this case, the position of anomalies [−1;1] and [1;−1] depends on the wavelength of light incident onto the structure. This behavior of the Wood anomalies can be explained by the fact that for the considered lattice the x- and y-directions are not equivalent. Namely, along the x-direction there exists the shortest distance between nodes, but not along the y-direction. Importantly, for Wood anomaly [1;1], the wavelength does not depend on deformation γ along the y axis ($\lambda_{y}\left(i=j=1\right)=const$). Note, for anomalies [1;0] and [0;1] (red dashed line in Figure 2), the dependences are different for the compressed and extended structures. In particular, at γ>1, this dependence is almost linear, while at γ<1, the dependence nonlinearity becomes noticeable. Moreover, this line shows that infinitely small deformations the sensitivity for compression and stretching is equal. However, for large deformation the difference becomes considerable. In terms of sensing, the case of straining the grating along the x axis is more attractive, as, in this case, one of Wood’s anomalies and, consequently, the resonance line position are independent of λ and can be used as the convenient reference.

For square array the Equations (Equation 10) and (Equation 12) can be rewritten as
(13)$$\lambda_{x}=\frac{a}{\sqrt{\left(\frac{1}{\gamma^{2}}i^{2}+j^{2}\right)}}n_{ext},\;\lambda_{y}=\frac{a}{\sqrt{\left(i^{2}+\frac{1}{\gamma^{2}}j^{2}\right)}}n_{ext}.$$
In these equations, the term for composition ij is equal to zero, due to the orthogonality of the direct- and reciprocal-lattice vectors and zero scalar product.

## 4. Results and Discussion

The calculated transmittance spectra are shown in Figure 1b. It can be seen that the initial triangular structure obtains D${}_{6h}$-symmetry and has one resonant line at a wavelength of λ=0.673 μm.

Note that irradiation of the D${}_{6h}$-symmetric structure by circularly polarized light does not lead to the splitting of the resonance line or a change in its wavelength. This is due to the symmetry of the structure, which causes the degeneracy of the resonance frequency. In a “live” experiment, the transmission line can appear much broader than the calculated one due to a decrease in the structural ordering and the impact of edge effects [28,30].

The symmetry violation in the structure due to the change in the grating period along the x axis leads to the degeneracy elimination and splitting of the resonance line. We calculated the transmittance spectra of the structure by the FDTD method (Figure 3a). It can be seen that the smooth extension along the x axis leads to the splitting of the transmission line. Note that the wavelength of the short-wavelength peak does not change, while the long-wavelength peak undergoes a red shift. In addition, this figure shows the positions of the resonance lines determined using Equation (Equation 10). The results obtained by the two independent methods are in excellent agreement. The resonance observed in the initial structure at a wavelength of λ=0.643 μm after lattice straining shifts to λ=0.943 μm. Note that the brightness for the short-wavelength line is much less than for the long-wavelength line. This effect can be explained by the fact that for the short-wavelength line the coupling with the incident field is not critical, while for the second line this critical coupling condition is satisfied. It is important that Equation (Equation 10) can describe the position of the resonance line upon further straining of the structure, which will lead us again to the D${}_{6h}$ symmetry. In this case, the resonance will be observed at a wavelength of λ=1.153 μm (Figure 3b). Note that at the D${}_{6h}$, D${}_{4h}$, and D${}_{{6h}^{\prime }}$ points, the structure has a high-order symmetry with degenerated modes, while in the intermediate states, the lattice is deformed to a lower-symmetric D${}_{2h}$ state and the mode degeneration is partially removed.

We believe that such a two-resonance structure can be used as an optical sensor or a detector, which makes it possible to determine the presence of strain and its value in the structure. The sensitivity of such a device can be defined as a wavelength difference between two resonances divided by the strain coefficient
(14)$$S=\frac{\lambda_{1}-\lambda_{2}}{(\gamma -1)\cdot 100\%}$$

Figure 4 presents the dependences of the wavelengths of the split peak upon compression or extension of the structure under study.

The calculation shows that the proposed model ensures a sensitivity of S=5.2 nm/% upon straining along the x axis, while the compression along this axis ensures a value of S=4.9 nm/% (see Figure 4a). The structure straining along the y axis ensures sensitivities of S=4.8 nm/% and S=5.1 nm/% (Figure 4b). Note that such a device is attractive because of the possibility of measuring a distance between two resonances, which leads to an increase in the accuracy, whereas in most similar devices the shift of one line is measured. Moreover, such sensitivity is only provided by the grating strain without introducing the shape anisotropy of constituent elements, which greatly simplifies the implementation of the device.

## 5. Conclusions

Thus, we studied the spectral properties of a two-dimensional array of dielectric nanodisks placed in a homogeneous medium. The analytical expressions for describing the evolution of Wood’s anomalies and, consequently, positions of the resonance lines upon straining the grating in two orthogonal directions were derived. It was found that, under straining the lattice along the x axis, the wavelength of only anomalies [1;0] and [0;1] change, while the wavelength of anomaly [1;1] remains invariable. However, for the structure strained along the y axis, positions of all three Wood’s anomalies exhibit the dependence on the wavelength of light incident onto the structure. These results were confirmed using finite-difference time-domain method. The possibility of using such structures as strain sensor was discussed. It was demonstrated that straining along the x axis ensures higher sensitivity than in the orthogonal direction. We believe that our results deepen the understanding of the behavior of Wood’s anomalies in the structures with symmetry violation and form a theoretical basis for creating strain sensors with the advanced spectral characteristics.

## Figures and Tables

**Figure 1 nanomaterials-11-01022-f001:**
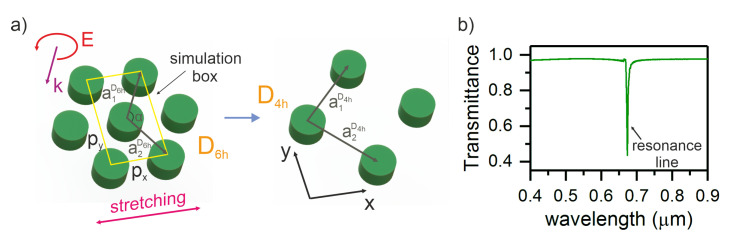
(**a**) Schematic of a two-dimensional triangular grating of nanodisks and (**b**) transmittance spectra of the structure calculated by the finite difference time domain method.

**Figure 2 nanomaterials-11-01022-f002:**
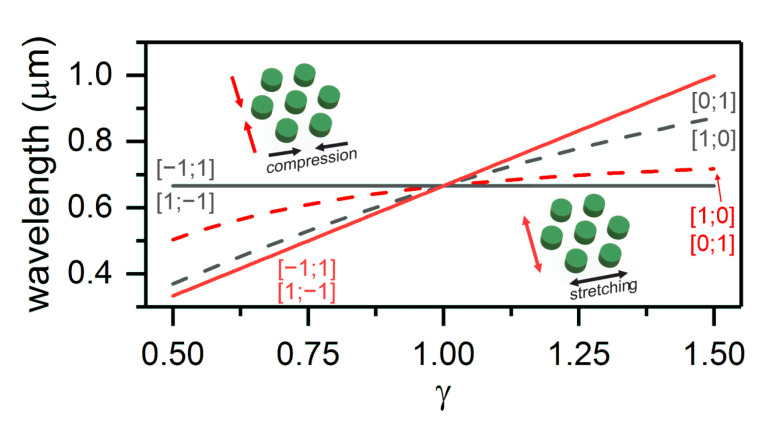
Positions of Wood’s anomalies at different coefficients γ of straining the structure along the *x* and *y* axes. The regions with γ<1 correspond to the compressed structure, and the regions with γ>1 correspond to the extended one. The notations [0;1], [1;0], and [−1;1], [1;−1] correspond to different values of the indices *i* and *j*.

**Figure 3 nanomaterials-11-01022-f003:**
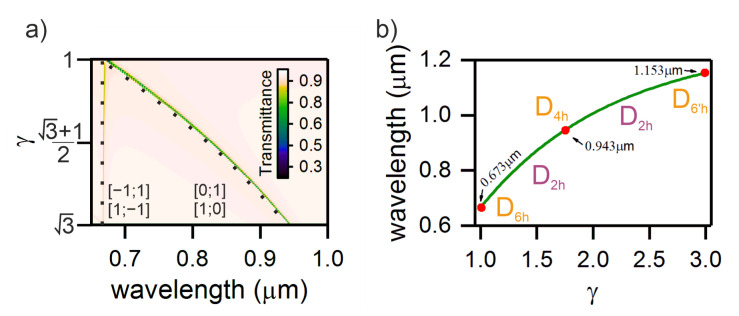
(**a**) Transmittance spectra of the two-dimensional structure with increasing $\gamma =p/p_{x}$. Closed circles show the positions of the resonance line calculated using Equation (Equation 10). (**b**) Dynamics of the resonance line wavelength during the transition of the D${}_{2h}$-symmetric structure between the high-symmetric points D${}_{6h}$, D${}_{4h}$, and $D_{6^{\prime }h}$.

**Figure 4 nanomaterials-11-01022-f004:**
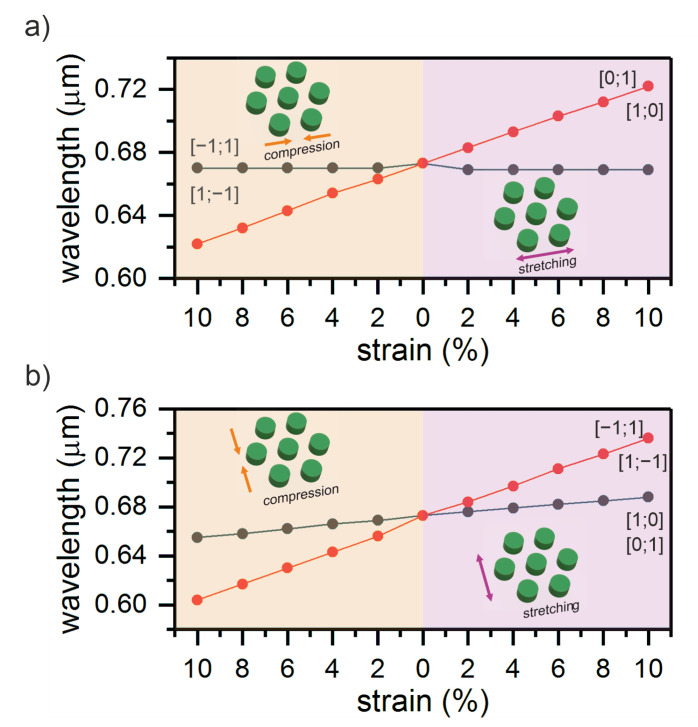
Dependences of the wavelengths of the split peak upon compression or extension of the structure along (**a**) the x and (**b**) y axes.

## Data Availability

The data presented in this study are available upon reasonable request from the corresponding author.

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
