# Peer review of "Strain Sensor via Wood Anomalies in 2D Dielectric Array"

_nanomaterials, 2021, doi:10.3390/nano11041022_

Round 1

Reviewer 1 Report

The authors of this work investigated, both analytically and numerically, the spectral properties of a two-dimensional array of dielectric nanodisks placed in a homogeneous medium. The analytical expressions for describing the Wood’s anomalies and the resonance lines in two orthogonal directions are obtained, showing that under straining the lattice along the x axis, positions of only anomalies [1;0] and  [0;1] change with the wavelength, while for the structure strained along the y axis, all the three Wood’s anomalies exhibit the dependence on the light wavelength.  This manuscript is well written and the results obtained are interesting and may have potential sensing applications in biomimetic optical nanomaterials. I have only two minor comments:

1) In page 2, line 58, the authors stated that the mathematical model underlying the FDTD (finite difference time domain ) method does not use approximations and is accurate for classical electrodynamics.  I agree with this. However, what mathematical model is adopted here? The authors should make some remarks on their model used.

2)  In page 3, line 92, what is the angle \theta used for the diffracting-wave vector? This parameter is not defined in the text. It seems that Eqs. (8) and (10) are obtained with \theta=0. Is it possible that similar results can be obtained with nonzero \theta?  Here, on lines 96 and 97, \alpha/2 should be enclosed by parentheses, to avoid confusion. Besides, in the caption of Fig. 2, the notations, e.g., [0;1], should be explicitly defined.

When the above issues are clarified, I recommend the publication of this paper in Nanomaterials.

Reviewer 2 Report

Bikbaev and coauthors report on a theoretical work describing the dependence of the optical properties of the lattice of dielectric disks on lattice deformations. The initial structure has hexagonal order and is either compressed or stretched. The shift of the Wood anomalies was calculated using FDTD. Such structures have been studied both theoretically and experimentally for some time. In this respect, the results of this work are not particularly groundbreaking. Nevertheless, I find the analytical description very illustrative. However, a major point of criticism is the lack of connection between the analytical description and the numerical calculations using FDTD. If the analytical model is correct, why are the FDTD calculations necessary? Shouldn't Eq. 8 and 10 be sufficient to describe the behavior of the anomalies? This needs to be much more clearly presented in the manuscript.  In summary, I consider the topic of the paper to be very well in line with the scope of the journal Nanomaterials. I am happy to support a publication, but certain inconsistencies and ambiguities should be clarified beforehand. My comments and questions should help the authors to revise their manuscript.

Comments:

  1. The title and abstract should clearly indicate that it is a theoretical work. Also, the term “biosensor” is not appropriate since it is simply a “strain sensor”. The “bio” aspect is not clear at all and thus I recommend not to focus on it.
  2. A major point of criticism is the lack of connection between the analytical description and the numerical calculations using FDTD. If the analytical model is correct, why are the FDTD calculations necessary? Shouldn't Eq. 8 and 10 be sufficient to describe the behavior of the anomalies? Please clarify and revise.
  3. In Figure 3a a faint red shadow is hardly visible. It seems that this color should indicate the transmittance as indicated by the color scale. However, it is very hard to see. Please revise.
  4. Figure 2: It is not clear why the y (red) and x (black) stretching/compression results in different position of anomalies. Since the excitation light is polarized circularly, the effect on the anomalies could be expected to be alike for both stretching/compression directions. In addition, the description in Lines 102-111 are not helpful to clarify this. This needs to be clarified in detail.
  5. The conclusions drawn in lines 94-97 are hard to follow and need further explanations. Please revise.
  6. Figure 4 is not clear. The color code (blue, green, red, purple) and the point styles (triangles, circles, squares) need to be explained.
  7. From my understanding the dependency of the wavelengths of the split peak upon compression and extension of the structure should be identical for both directions (Figure 4a and b). I recommend to double check the FDTD simulations for errors.
  8. The refractive index environment is assumed to be homogenous, for the sake of simplicity. However, it is not clear why a medium index of 1.33, which corresponds to water, was assumed in the model. Considering a flexible support (e.g. made of PDMS) its refractive index would range probably between 1.45 and 1.55, which is very close to the refractive index used for the disks. In addition, the importance of the medium homogeneity needs to be discussed (see Ponomareva et al. DOI: 10.1021/acs.langmuir.0c02430)
  9. It is not clear why the wavelength of anomaly [1;1] remains invariable. From several experimental work, mode splitting was observed, this is a blue-shift of the compressed mode and a red-shift of the expanded mode (see Charconnet et al. DOI: 10.1109/OMN.2019.8925038). This is especially the case for asymmetric lattice deformations, which take place during the stretching of a flexible support such as PDMS (with a Poisson’s ration of approx. 0.5). Thus, an extension of the support is accompanied by a compression in perpendicular direction. This needs to be discussed in detail due to its high importance to experimental studies.
  10. It is not clear why the two-dimension grating is described as triangular. Isn’t it of hexagonal order?
  11. How would Eq.8 and 10 change for a square lattice. I am confident that many readers would be interested in this aspect. Including this aspect would improve the significance of this work.
  12. Line 45: “… of nanoparticles…” should be “… of nanodisks…”.
  13. The difference between symmetric and asymmetric lattice deformation needs to be discussed in more detail. Please revise.
  14. In line 28, various experimentally studied lattices are mentioned such as spheroids, disks, dimers, or large 3D clusters. However, between simple dimers and complex 3D clusters, also “flat/2D” clusters have attracted a lot of interest, e.g. see heptamers as studied by Charconnet et al. (DOI: 10.1109/OMN.2019.8925038). The authors might wish to complete their list with examples of “flat/2D” clusters.

Round 2

Reviewer 2 Report

The authors have revised their contribution. I welcome the changes to the title (removal of the prefix "bio") and the expanded discussion in the main text. The consideration of square lattices is very helpful. The revisions of the figures seems appropriate. The only drawback is that the title is still somewhat misleading and does not clearly indicate that this is a theoretical work. I recommend correcting this and to clearly mention in both the title and abstract that this is a theoretical work. In the abstract, the authors might wish also to indicate that both Mit theory and FDTD simulations have been employed. Apart from these minor revisions, I recommend publication.  
